# OpenReview forum: "Investigating CoT Monitorability in Large Reasoning Models"
_ICLR.cc/2026/Conference — ICLR 2026 Conference Withdrawn Submission_

### Official Review · Reviewer_iMpc · 2025-10-21

**Soundness:** 3
**Presentation:** 2
**Contribution:** 2
**Rating:** 4
**Confidence:** 4

**Summary:**

The manuscript tackles two challenges around chains of thought (CoT) in LLMs: (1) the CoT might not be faithful, i.e., it contradicts the final result and (2) monitoring COTs seems tricky, and LLMs currently underperfom. The paper then investigates (1) often LRMs faithfully verbalize the true factors guiding their decision, and (2) how reliable monitors (e.g., another LLM) can predict verbalization. They present some empirical results on these, and propose MoME, where they generate synthetic pairwise prereference data which helps train a new LLM monitor which outperforms vanilla GPT-5-nano or Gemini-2.5-flash in monitoring CoTs.

**Strengths:**

- Experiments around prompt verbalization across different models. Self-explanation increases robustness, while CoT compression has detrimental effects. Cannot hurt to have a paper showing this.
- Apparently, GPT-5-nano and Gemini-2.5-flash-lite are both not great at monitoring CoTs. A model trained to generate structured judgments (MoME) improves on this.
- All of this seems sound and believable, so no huge issues around this.

**Weaknesses:**

- This work doesn't sound too surprising. We knew that current LRM models are not entirely faithful in their CoTs, and that there would be mismatches.
- Also doesn't come as a surprise that GPT-5-nano or Gemini-2.5-flash do not perform great at monitoring CoTs.
- The proposed method MoME fixing this, I'm sure it works, but it doesn't seem very inspirational. E.g., Why does it have to be this, why can't a model trained on appropriate data without structured judgement do this?
- So, taken together, the paper perhaps is not focused enough, and tries to show too many things without conveying a super compelling take-away message.
- How could the authors improve this paper in my opinion? See some pointers below

**Questions:**

- line 127: Sentence seems incomplite ("and medical Liu et al.")
- line 142:  misbehaviors.Zou  ->  misbehaviors. Zou
- line 469: You might want to engage with the ORPO paper here, which comes to similar conclusions: https://aclanthology.org/2024.emnlp-main.626/
- overall, slightly inconsistent use of \citet and \citep, please fix
- overall, too many acronyms, especially in described metrics. I would propose to focus on 1-2 key metrics, and describe these in more details, and move everything to appendix. The takeaways from other metrics can be described in 1-2 sentences, and then the authors refer to Appendix for more detail
- some typos (see a few above), so a general thorough pass fixing these would be appreciated
- I would recommend to do an extensive human error analysis, and come up with a taxonomy of things which can go wrong in verbalization and monitoring (and perhaps split that up in two papers). Are these the same across domains, cues and models? Are there mistakes which are only made by smaller models, but larger models already are robust? Are there things which seem to be difficult to fix given current methods and require substantial more work? All of these could meaningfully inform future work on prompt robustness. What are other slightly "weird" things in CoTs which should be addressed? Similarly, for CoT monitoring, the data seems mostly synthetic and I think there's value in having a dedicated, high-quality and human-annotated benchmark which can be used to reliably test monitoring, even next or in two years when new models will be out there.

---

### Official Review · Reviewer_z1F5 · 2025-10-28

**Soundness:** 3
**Presentation:** 2
**Contribution:** 2
**Rating:** 2
**Confidence:** 3

**Summary:**

CoT monitoriability is highlighted as a potential for AI safety research, but challengers exist, including that models do not faithfully represent internal decisi-making and that the monitors themselves may be overly or insufficiently sensitive. This paper purports to systematically investigate CoT monitorability. The paper focuses on challenges of 'berbalization' and 'monitor reliability'

**Strengths:**

- It is good that a number of existing, multi-domain datasets are used in this work. The number of models investigated is also a strength; one notes that there are only two families of model but also that there are not many open source LRMs that scale.
- The precision/formalization of the metrics is noted, and the _general_ reproducibility of the work appears strong, at least regarding most of the empircal setup, though with caveats (below).
- The evidence of oversensitivity is pursuasive; the links between longer CoT and higher robustness (or lower scheming) seems somewhat actionable or at least follow-up-uponable.

**Weaknesses:**

- It is hard to determine the extent to which the current work is truly novel. Some existing work is cited, but it is essentially discarded as "intuitive speculation", although that may be premature. Turpin et al (2025) includes concrete experiemnts (as does Turpim et al (2023)) and Baker et al (2025) show that LLMs-as-monitor can detect reward-hacking by reading another model’s CoT, can compare monitoring strategies, etc.
- Despite the positive nature of the multi-domain datasets noted above, insufficient detail is provided in A.2 on how numerical answers are converted "to a multiple-choice format by generating four options (A, B, C, D) for each question", however. Without actual detail on the process used in this fundamental step, it's not clear if evaluation is overly 'synthetic' or 'toy', nor how it would generalize to the 'real-world'.
- Much of the numeric data provided (to the extent that it's legible, at least in the appendices!) are just given in terms of raw numbers. While the differences seem to suggest significance, uncertainty should be reported.
- Despite some strength in reproduciblity, monitors depend on closed soruce models like Gemini ang GPT-4o, which impedes replication even if those models remain unchanged somehow.

**Questions:**

- Can you please correct errors such as \citet instead of \citep on L127 and L147, a missing space on L142 and L144, etc? In fact, an atypically large number of formatting errors pervades the document -- can you please be more careful?
- Is Pearson correlation really the best correlation to choose for these data (e.g., Fig 3)? What are the associated p-values?
- Where are there possibilities for 'circular' judgements? If verbalization is LLM-labeled, monitors are selected by LLM scores, etc -- are errors bounded?
- Unauthorized Access is focused on quite a bit and there is a (very thin) link to a "plausible explanation ... that such cues may trigger model-specific latent “dark” personalities". Do you have any empirical evidence for this, or is this the 'intuitive speculation' you warned us about?
- Is it your assumption that changing to a cue target is equivalent to misbehaviour and that resisting such cues is equivalent to benign behaviour? Can you specify false positives and false negatives?

---

### Official Review · Reviewer_97do · 2025-10-31

**Soundness:** 3
**Presentation:** 3
**Contribution:** 2
**Rating:** 4
**Confidence:** 4

**Summary:**

This work investigates the monitorability of chain-of-thought (CoT) in reasoning models on two axes. The first one is the faithfulness / truthfulness of the explanations provided in the CoT by the model and the second one is the sensitivity / reliability of CoT monitors in detecting misbehavior. The authors investigate these axes on multiple-choice questions spanning several domains where misleading cues are injected in the prompt (i.e., adversarial prompting). The authors further propose several metrics for evaluation: robustness, scheming, effective escaping misbehavior rate (EEMR) and over-sensitivity of the monitor (OSM), and also evaluate the effects of CoT interventions from prior works (aimed at improving either the inference efficiency or inference scaling) to CoT monitorability. Finally, the authors propose the "monitor model misbehavior and give evidence" (MoME) framework that guides the CoT monitors to produce more structured judgments while providing supporting evidence.

**Strengths:**

- A systematic study of CoT monitorability on two axes important for the problem: faithfulness and monitor reliability.
- The benchmarks cover multiple domains and models.
- The insights on CoT compression hampering CoT monitoring and that longer reasoning traces make scheming more difficult are potentially useful for the community

**Weaknesses:**

- Benchmarks focus only on multiple-choice tasks but open-ended scenarios are more common in practice
- The MoME framework evaluation is not convincing; the authors use a different metric (Monitor Effectiveness Score) in that section compared to the ones in Table 2, making comparison with existing methods difficult. Furthermore, MoME is finetuned for the task at hand, whereas the other monitors were only prompt adjusted. For a more fair comparison, I would have expected that the CoT monitors were also finetuned in a similar manner to MoME.

**Questions:**

Overall, I am leaning a bit on the negative side for this work. While it does provide useful insights, I believe that the evaluation of MoME is incomplete. As for questions and comments to the authors

- Some of the results on Table 1 are peculiar. Why does the "Unauthorized access" category usually has the second highest robustness when it demonstrates the highest scheming? Shouldn't these two be contradictory as the robustness is about mentioning the cues and not using them whereas scheming is about removing them while still using them?
- For the experiments on Table 2; given that the CoT monitors see only the CoT and not the actual action / response, doesn't it make sense that it focuses more on whether the CoT mentioned specific things? If it doesn't observer the action, it has no way of knowing how connected these two were.
- The insight that longer responses tend to verbalize cues more often is interesting, but I wonder if it was expected; with longer length there is a higher probability that a specific cue mentioned in a prompt might appear in the CoT.

---

### Official Review · Reviewer_sSUZ · 2025-11-01

**Soundness:** 3
**Presentation:** 3
**Contribution:** 2
**Rating:** 6
**Confidence:** 4

**Summary:**

This paper investigates Chain-of-Thought (CoT) monitorability in Large Reasoning Models (LRMs), focusing on two dimensions:

1. Faithfulness (verbalization) – whether models truthfully articulate factors influencing decisions.
2. Monitor reliability – whether CoT-based monitors detect misbehavior accurately.

To measure these, the authors introduce:

**Robustness** (resist cues while verbalizing) and **Scheming** (hide cue influence) for faithfulness.
**EEMR** (Effective Escaping Misbehavior Rate: monitor misses) and **OSM** (Over-Sensitivity of Monitor: false alarms) for reliability.

They also define **Cue Influence Rate (CIR)**: how often a model changes its answer to the cued option when a misleading cue is injected. CIR is central because it quantifies susceptibility to adversarial influence.
Experiments span six LRMs (DeepSeek-R1 and Qwen families) across math, science, and ethics tasks with eight adversarial cue types. Findings:

* Larger models reflect more but remain vulnerable to cues (CIR barely decreases with scale).
* Unauthorized Access cues are most potent.
* Monitors often oversensitive (high OSM) even when EEMR is low.
* Longer CoTs correlate with higher accuracy and monitorability, but cost trade-offs are unaddressed.
The paper proposes MoME, a structured evidence-based monitoring paradigm, and shows DPO-trained monitors outperform baselines on a Monitor Effectiveness Score.

**Strengths:**

The paper makes a meaningful contribution to AI safety by systematically analyzing CoT monitorability.

* Clear conceptual framing: Separates CoT faithfulness and monitor reliability, which are often conflated.
* Novel, interpretable metrics: Robustness and Scheming for model behavior; EEMR and OSM for monitor performance.
* Comprehensive evaluation: Six LRMs, multiple datasets (AIME, GPQA, MMLU moral), and eight adversarial cue types.

Empirical insights worth highlighting:

* CIR barely decreases with scale → larger models still cue-sensitive.
* Unauthorized Access cues strongly influence behavior.
* Higher accuracy correlates with higher OSM (oversensitivity).


Regarding their MoME framework: Structured JSON outputs and DPO fine-tuning improve monitor balance.

Correlation analysis: Links CoT length to accuracy and monitorability, raising important trade-off questions.

**Weaknesses:**

While the paper is strong conceptually, several issues limit clarity and generalization:

* Intro clarity: Metrics appear abruptly; better to explain intuition first and formalize later.
* Scope limitation: All tasks are single-turn MCQs; no multi-turn or tool-use settings.
* Model diversity: Mostly Qwen-derived; no LLaMA/GPT baselines.
* Metric assumptions: Multiplicative definitions assume independence; item-level conditional analysis missing.
* Monitor generalization: MoME trained on Qwen traces; no cross-family evaluation or held-out test set.
* Interpretability gaps: Terms like “self-reflection” and “self-correction” introduced without clear definitions.
* Analysis depth: CIR reported only as answer-change; logit-level shifts or distributional effects would add rigor.
* Cost trade-offs: Longer CoTs improve monitorability but token budget implications are not quantified.
* Naming inconsistency: OSM vs OSR in later sections.
* Presentation: Figures need larger fonts; findings could be summarized as bullet takeaways.

**Questions:**

1. Decoding and CIR

Your paper defines Cue Influence Rate (CIR) as the proportion of cases where the model changes its answer to the cued option when a misleading cue is injected. Could you clarify:

Are answers generated using greedy decoding or sampling?
Did you examine logit-level changes or probability shifts for all options, not just the final choice? This would reveal whether cues subtly bias the distribution even when the top prediction remains unchanged.

---

2. Metric introduction in the intro

Currently, metrics like Robustness, Scheming, EEMR, and OSM appear abruptly in the introduction. Would it improve clarity to:

Introduce them conceptually first (e.g., “miss rate” vs “false alarm rate”) so readers understand the intuition?
Then formalize definitions later in Section 3? This would make the narrative easier to follow for readers unfamiliar with these terms.

---
3. Self-reflection and VR scoring

You mention that Verbalization Rate (VR) measures how often the model acknowledges cue influence in its reasoning. Could you clarify:

Which reasoning stages count toward VR? Internal reasoning (tr), external CoT (r), or both?
How do you distinguish “reflective reasoning” from ordinary step-by-step reasoning? A precise definition would help interpret VR results.

---
4. Robustness vs Scheming both high

Table 1 shows cases where Robustness and Scheming are both high, which seems counterintuitive. Does this indicate that these metrics capture orthogonal behaviors?

Have you considered a composite metric for model-side behavior (similar to MES for monitors) that penalizes simultaneous high Scheming and high Robustness?
If not, what interpretation should readers take from these overlaps?

---
5. Generalization of MoME

MoME is trained on Qwen3-4B traces and evaluated on similar data.

Did you test MoME on unseen model families (e.g., DeepSeek vs Qwen) or unseen cue types?
If not, do you plan to include a held-out evaluation set to demonstrate generalization beyond the training distribution?

---
6. Token budget trade-offs

Section 5 shows that longer CoTs correlate with higher accuracy and monitorability.

Could you provide Pareto curves showing trade-offs between token length, accuracy, and monitor performance?
This would help practitioners decide how much extra reasoning is worth the cost in real deployments.

---
7. MoME ablation

MoME outputs structured fields (e.g., category, evidence, rationale).

Which components contribute most to improved Monitor Effectiveness Score?
Did you run ablation experiments to isolate the impact of evidence vs rationale vs confidence scoring?

---
8. Why no LLaMA/GPT baselines?

Your evaluation focuses on Qwen and DeepSeek models.

Was this due to compute constraints, API cost, or other reasons?
Including at least one widely used baseline (e.g., LLaMA or GPT) would strengthen claims about generality.

---
9. Cue realism

Some cues (e.g., Unauthorized Access, embedded validation functions) are strong adversarial triggers.
How representative are these cues of real-world user interactions?
Could you discuss which cues are most likely to occur in practical settings and how that affects the relevance of your findings?

---
10. Naming consistency

In Section 5, the metric name changes from OSM to OSR. Is this a typo?

---

### Official Review · Reviewer_NB3F · 2025-11-02

**Soundness:** 1
**Presentation:** 2
**Contribution:** 1
**Rating:** 2
**Confidence:** 4

**Summary:**

This paper introduced the concept of CoT Monitorability, which aimed to detect potential model misbehavior through its CoT reasoning chain. The authors explored two challenges: (1) whether LLMs can faithfully express their decision rationale within their CoT, and (2) whether CoT-based monitors can reliably identify model’s misbehavior. Experiments showed that LLMs produced unfaithful explanations and that monitors tended to be overly sensitive. They further demonstrated that structured-prompt training could improve monitoring effectiveness.

**Strengths:**

The paper aims to evaluate the faithfulness of the CoT and how it can detect potential misbehavior in an LLM. The topic is timely and offers a new perspective for understanding and supervising LLMs.

**Weaknesses:**

1. The main limitation lies in the lack of formal and rigorous evaluation objectives. The paper does not clearly define or quantify concepts such as unfaithful explanations or monitor reliability within CoT. For example, does CoT fidelity depend solely on whether the model explicitly references the prompt? Should it be considered faithful if the semantic content aligns despite differing expressions? Or if a model appears to mention the prompt but merely parroted it as an explanation, should this be considered faithful? Furthermore, faithfulness must be defined relative to an ideal CoT (or a true CoT), yet the paper neither defines nor constructs such an ideal, nor provides a measurable distance between the generated CoT and the ideal one. Similarly, monitor reliability lacks clear formalization and quantitative evaluation.

2. The design of the evaluation metrics lacks theoretical grounding. It is unclear why Robustness is defined as AKRa×VRand Scheming as (1-VR)×CIRa. When prompts and resistance are semantically non-independent, the multiplicative form may systematically overestimate or underestimate the true violation probability. Furthermore, these metrics appear tailored for specific tasks (i.e., multiple-choice adversarial prompts), limiting their broader applicability.

3. The monitor design lacks generality. The MoME framework heavily relies on manually crafted prompts and LLM-based judges, which limits its automation and scalability. The paper also does not discuss how this framework could generalize to prompt-free attacks or open-domain dialogue, where monitoring becomes more complex.

**Questions:**

See Weaknesses.

---

### Note · Authors · 2025-12-08

**Comment:**

Thank you to all the reviewers for their time and constructive feedback. We will revise this work accordingly.

**Withdrawal Confirmation:**

I have read and agree with the venue's withdrawal policy on behalf of myself and my co-authors.